# Protein contact prediction from amino acid co-evolution using convolutional networks for graph-valued images

**Vladimir Golkov**[1], **Marcin J. Skwark**[2], **Antonij Golkov**[3], **Alexey Dosovitskiy**[4],
**Thomas Brox**[4], **Jens Meiler**[2], **and Daniel Cremers**[1]

[1] Technical University of Munich, Germany
[2] Vanderbilt University, Nashville, TN, USA
[3] University of Augsburg, Germany
[4] University of Freiburg, Germany

`golkov@cs.tum.edu, marcin@skwark.pl, antonij.golkov@student.uni-augsburg.de,`
`{dosovits,brox}@cs.uni-freiburg.de, jens.meiler@vanderbilt.edu, cremers@tum.de`

## Abstract

Proteins are responsible for most of the functions in life, and thus are the central focus of many areas of biomedicine. Protein structure is strongly related to protein function, but is difficult to elucidate experimentally, therefore computational structure prediction is a crucial task on the way to solve many biological questions. A contact map is a compact representation of the three-dimensional structure of a protein via the pairwise contacts between the amino acids constituting the protein. We use a convolutional network to calculate protein contact maps from detailed evolutionary coupling statistics between positions in the protein sequence. The input to the network has an image-like structure amenable to convolutions, but every "pixel" instead of color channels contains a bipartite undirected edge-weighted graph. We propose several methods for treating such "graph-valued images" in a convolutional network. The proposed method outperforms state-of-the-art methods by a considerable margin.

## 1   Introduction

Proteins perform most of the functions in the cells of living organisms, acting as enzymes to perform complex chemical reactions, recognizing foreign particles, conducting signals, and building cell scaffolds – to name just a few. Their function is dictated by their three-dimensional structure, which can be quite involved, despite the fact that proteins are linear polymers composed of only 20 different types of amino acids. The sequence of amino acids dictates the three-dimensional structure and related proteins share both structure and function. Predicting protein structure from amino acid sequence remains a problem that is still largely unsolved.

### 1.1   Protein structure and contact maps

The *primary structure* of a protein refers to the linear sequence of the amino acid residues that constitute the protein, as encoded by the corresponding gene. During or after its biosynthesis, a protein spatially folds into an energetically favourable conformation. Locally it folds into so-called *secondary structure* ($\alpha$-helices and $\beta$-strands). The global three-dimensional structure into which the entire protein folds is referred to as the *tertiary structure*. Fig. 1a depicts the tertiary structure of a protein consisting of several $\alpha$-helices.

Protein structure is mediated and stabilized by series of weak interactions (physical contacts) between pairs of its amino acids. Let $L$ be the length of the sequence of a protein (i.e. the number of its amino acids). The tertiary structure can be partially summarized as a so-called *contact map* – a sparse $L \times L$ matrix $C$ encoding the presence or absence of physical contact between all pairs of $L$ amino acid residues of a protein. The entry $C_{i,j}$ is equal to $1$ if residues $i$ and $j$ are in contact and $0$ if they are not. Intermediate values may encode different levels of contact likeliness.

We use these intermediate values without rounding where possible because they hold additional information. The "contact likeliness" is a knowledge-based function derived from Protein Data Bank, dependent on the distance between $C\beta$ atoms of involved amino acids and their type. It has been parametrized based on the amino acids' heavy atoms making biophysically feasible contact in experimentally determined structures.

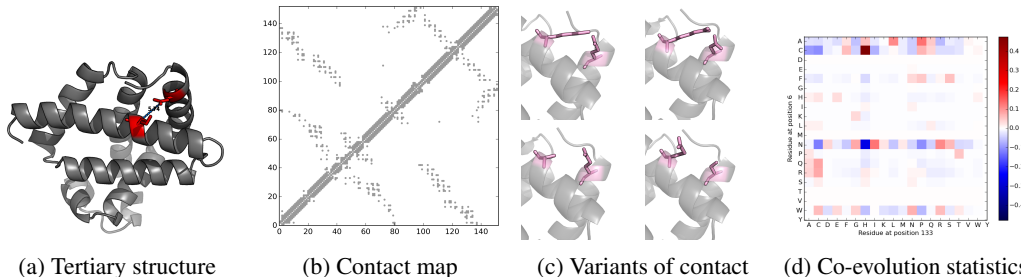

| (a) Tertiary structure | (b) Contact map | (c) Variants of contact | (d) Co-evolution statistics |

Figure 1: Oxymyoglobin (a) and its contact between amino acid residue 6 and 133. Helix–helix contacts correspond to "checkerboard" patterns in the contact map (b). Various variants of the contact 6/133 encountered in nature (native pose in upper left, remaining poses are theoretical models) (c) are reflected in the co-evolution statistics (d).

## 2 Methods

The proposed method is based on inferring direct co-evolutionary couplings between pairs of amino acids of a protein, and predicting the contact map from them using a convolutional neural network.

### 2.1 Multiple sequence alignments

As of today the UniProt Archive (UniParc [1]) consists of approximately 130 million different protein sequences. This is only a small fraction of all the protein sequences existing on Earth, whose number is estimated to be on the order of $10^{10}$ to $10^{12}$ [2]. Despite this abundance, there exist only about $10^5$ sequence families, which in turn adopt one of about $10^4$ folds [2]. This is due to the fact that homologous proteins (proteins originating from common ancestors) are similar in terms of their structure and function. Homologs are under evolutionary pressure to maintain the structure and function of the ancestral protein, while at the same time adapting to the changes in the environment.

Evolutionarily related proteins can be identified by means of homology search using dynamic programming, hidden Markov models, and other statistical models, which group homologous proteins into so-called *multiple sequence alignments*. A multiple sequence alignment consists of sequences of related proteins, aligned such that corresponding amino acids share the same position (column). The 20 amino acid types are represented by the letters A,C,D,E,F,G,H,I,K,L,M,N,P,Q,R,S,T,V,W,Y. Besides, a "gap" (represented as "–") is used as a 21st character to account for insertions and deletions.

For the purpose of this work, all the input alignments have been generated with jackhmmer, part of HMMER package (version 3.1b2, http://hmmer.org) run against the UniParc database released in summer 2015. The alignment has been constructed with the E-value inclusion threshold of 1, allowing for inclusion of distant homologs, at a risk of contaminating the alignment with potentially evolutionarily unrelated sequences. The resultant multiple sequence alignments have not been modified in any way, except for removal of inserts (positions that were not present in the protein sequence of interest). Notably, contrary to many evolutionary approaches, we did *not* remove columns that (a) contained many gaps, (b) were too diverse or (c) were too conserved. In so doing, we emulated a fully automated prediction regime.

## 2.2 Potts model for co-evolution of amino acid residues

Protein structure is stabilized by series of contacts: weak, favourable interactions between amino acids adjacent in space (but not necessarily in sequence). If an amino acid becomes mutated in the course of evolution, breaking a favourable contact, there is an evolutionary pressure for a compensating mutation to occur in the interacting partner(s) to restore the protein to an unfrustrated state. These pressures lead to amino acid pairs varying in tandem in the multiple sequence alignments. The observed covariances can subsequently be used to predict which of the positions in the protein sequence are close together in space.

The directly observed covariances are by themselves a poor predictor of inter-residue contact. This is due to transitivity of correlations in multiple sequence alignments. When an amino acid *A* that is in contact with amino acids *B* and *C* mutates to *A'*, it exerts a pressure for *B* and *C* to adopt to this mutation, leading to a spurious, indirect correlation between *B* and *C*. Oftentimes these spurious correlations are more prominent than the actual, direct ones. This problem can be modelled in terms of one- and two-body interactions, analogous to the Ising model of statistical mechanics (or its generalization – the Potts model). Solving an inverse Ising/Potts problem (inferring direct causes from a set of observations), while not feasible analytically, can be accomplished by approximate, numerical algorithms. Such approaches have been recently successfully applied to the problem of protein contact prediction [3, 4].

One of the most widely-adopted approaches to this problem is pseudolikelihood maximization for inferring an inverse Potts model (plmDCA [3, 5]). It results in an $L \times L \times 21 \times 21$ array of inferred evolutionary couplings between pairs of the $L$ positions in the protein, described in terms of $21 \times 21$ coupling matrices. These coupling matrices depict the strength of evolutionary pressure at particular amino acid type pairs (e.g. histidine–threonine) to be present at this position pair – the higher the value, the more pressure there is. These values are not directly interpretable, as they depend on the environment the amino acids are in, their propensity to mutate and many other factors. So far, the best approach to obtain scores corresponding to contact propensities was to compute the Frobenius norm of individual coupling matrices rendering a contact matrix, which then has been subject to average product correction [6]. Average product correction scales the value of contact propensity based on the mean values for involved positions and a mean value for the entire contact matrix.

As there is insufficient data to conclusively infer all the parameters, and coupling inference is inherently ill-posed, regularization is required [3, 5]. Here we used $l_2$ regularization with $\lambda = 0.01$.

These approaches to reduce each $21 \times 21$ coupling matrix to only one value discard valuable information encoded in matrices, consequently leading to a reduction in expected predictive capability. In this work we use the entire $L \times L \times 21 \times 21$ coupling data $\mathbf{J}$ in their unmodified form. The value $\mathbf{J}_{i,j,k,l}$ quantifies the co-evolution of residue type $k$ at location $i$ with residue type $l$ at location $j$. The $L \times L \times 21 \times 21$ array $\mathbf{J}$ serves as the main input to the convolutional network to predict the $L \times L$ contact map $C$.

The following symmetries hold: $C_{i,j} = C_{j,i}$ and $\mathbf{J}_{i,j,k,l} = \mathbf{J}_{j,i,l,k} \forall i, j, k, l$.

## 2.3 Convolutional neural network for contact prediction

The goal of this work is to predict the contact $C_{i,j}$ between residues $i$ and $j$ from the co-evolution statistics $\mathbf{J}_{i,j,k,l}$ obtained from pseudolikelihood maximization [3]. Not only the local statistics $(\mathbf{J}_{i,j,k,l})_{k,l}$ for fixed $(i,j)$ but also the neighborhood around $(i,j)$ is informative for contact determination. Particularly, contacts between different secondary structure elements are reflected both in the spatial contact pattern, such as the "checkerboard" pattern typical for helix–helix contacts, cf. Fig. 1b (the "$i$" and "$j$" dimensions), as well as in the residue types (the "$k$" and "$l$" dimensions) at $(i,j)$ and in its neighborhood. Thus, a convolutional neural network [7] with convolutions over $(i,j)$, i.e. learning the transformation to be applied to all $w \times w \times 21 \times 21$ windows of $(\mathbf{J}_{i,j,k,l})$, is a highly appropriate method for prediction of $C_{i,j}$.

The features in each "pixel" $(i,j)$ are the entries of the $21 \times 21$ co-evolution statistics $(\mathbf{J}_{i,j,k,l})_{k,l \in \{1,...,21\}}$ between amino acid residues $i$ and $j$. Fig. 1d shows the co-evolution statistics of residues 6 and 133, i.e. $(\mathbf{J}_{6,133,k,l})_{k,l \in \{1,...,21\}}$, of oxymyoglobin. These $21 \cdot 21$ entries can be vectorized to constitute the feature vector of length 441 at the respective "pixel".

The neural network input $\mathbf{J}$ and at its output $C$ should have the same size along the convolution dimensions "$i$" and "$j$". In order to achieve this, the input boundaries are padded accordingly (i.e. by the receptive window size) along these dimensions. In order to help the network distinguish the padding values (e.g. zeros) from valid co-evolution values, the indicator function of the valid region (1 in the valid $L \times L$ region and 0 in the padded region) is introduced as an additional feature channel.

Our method is based on pseudolikelihood maximization [3] and convolutional networks, *plmConv* for short.

## 2.4 Convolutional neural network for bipartite-graph-valued images

The fixed order of the 441 features can be considered acceptable since any input–output mapping can in principle be learned, assuming we have sufficient training data (and an appropriate network architecture). However, if the amount of training data is limited then a better-structured, more compact representation might be of great advantage as opposed to requiring to see most of the possible configurations of co-evolution. Such more compact representations can be obtained by relaxing the knowledge of the identities of the amino acid residues, as described in the following.

The features at "pixel" $(i, j)$ correspond to the weights of a (complete) bipartite undirected edge-weighted graph $K_{21,21}$ with $21 + 21$ vertices, with the first disjoint set of 21 vertices representing the 21 amino acid types at position $i$, the second set representing the 21 amino acid types at position $j$, and the edge weights representing co-evolution of the respective variants. Thus, $B = (\mathbf{J}_{i,j,k,l})_{k,l \in \{1,\ldots,21\}}$ is the biadjacency matrix of this graph, i.e. $A = \begin{pmatrix} 0 & B \\ B^T & 0 \end{pmatrix}$ is its adjacency matrix. The edge weights (i.e. entries of $B$) are different at each "pixel" $(i, j)$.

There are different possibilities of passing these features (the entries of $B$) to a convolutional network. We propose and evaluate the following possibilities to construct the feature vector at pixel $(i, j)$:

1. Vectorize $B$, maintaining the order of the amino acid types;
2. Sort the vectorized matrix $B$;
3. Sort the rows of $B$ by their row-wise norm, then vectorize;
4. Construct a histogram of the entries of $B$.

While the first method maintains the order of amino acid types, all others produce feature vectors that are invariant to permutations of the amino acid types.

## 2.5 Generalization to arbitrary graphs

In other applications to graph-valued images with general (not necessarily bipartite) graphs, similar transformations as above can be applied to the adjacency matrix $A$. An additional useful property is the special role of the diagonal of $A$. Node weights can be included as additional features, and accordingly reordered.

There has been work on neural networks which can process functions defined on graphs [8, 9, 10, 11]. In contrast to these approaches, in our case the input is defined on a regular grid, but the value of the input at each location is a graph.

## 2.6 Data sets

The Critical Assessment of Techniques for Protein Structure Prediction (CASP) is a bi-annual community-wide experiment in blind prediction of previously unknown protein structures. The prediction targets vary in difficulty, with some having a structure of homologous proteins already deposited in the Protein Data Bank (PDB), considered easy targets, some having no detectable homologs in PDB (hard targets), and some having entirely new folds (free modelling targets). The protein targets vary also in terms of available sequence homologs, which can range from only a few sequences to hundreds of thousands.

We posit that the method we propose is robust and general. To illustrate its performance, we have intentionally trained it on a limited set of proteins originating from CASP9 and CASP10 experiments

and tested it on CASP11 proteins. In so doing, we emulated the conditions of a real-life structure prediction experiment.

The proteins from these experiments form a suitable data set for this analysis, as they (a) are varied in terms of structure and "difficulty", (b) have previously unknown structures, which have been subsequently made public, (c) are timestamped and (d) they have been subject to contact prediction attempts by other groups whose results are publicly available. Therefore, training on CASP9 and CASP10 data sets allowed us to avoid cross-contamination. We are reasonably confident that any performance of the method originates from the method's strengths and is not a result of overfitting.

The training has been conducted on a subset of 231 proteins from CASP9 and CASP10, while the test set consisted of 89 proteins from CASP11 (all non-cancelled targets). Several proteins have been excluded from the training set for technical reasons: lack of any detectable homologs, too many homologs detected, or lack of structure known at the time of publishing of CASP sets. The problems with the number of sequences can be alleviated by attempting different homology detection strategies, which we did not do, as we wanted to keep the analysis homogeneous.

## 2.7 Neural network architecture

Deep learning has strong advantages over handcrafted processing pipelines and is setting new performance records and bringing new insights in the biomedical community [12, 13]. However, parts of the community are adopting deep learning with certain hesitation, even in areas where it is essential for scientific progress. One of the main objections is a belief that the craft of network architecture design and the network internals cannot be scientifically comprehended and lack theoretical underpinnings. This is a false belief. There are scientific results to the contrary, concerning the loss function [14] and network internals [15].

In the present work, we design the network architecture based on our knowledge of which features might be meaningful for the network to extract, and how.

The first layer learns 128 filters of size $1 \times 1$. Thus, 441 input features are compressed to 128 learned features. This compression enforces the grouping of similar amino acids by their properties. Examples of important properties are hydrophobicity, polarity, and size. Some of the most relevant parts of the input information "cysteine (C) at position $i$ has a strongly positive evolutionary coupling with histidine (H) at position $j$" (cf. Fig. 1d) is that the amino acids co-evolving have certain hydrophilicity properties; that both are polar; and that the one at position $i$ is rather small and the one at position $j$ is rather large; etc. One layer is sufficient to perform such a transformation. Note that we do not handcraft these features; the network learns feature extractors that are optimal in terms of the training data. Besides, compressing the inputs in this optimal way also reduces the number of weights of the subsequent layer, thus regularizing the model in a natural way, and reducing the run time and memory requirements.

The second layer learns 64 filters of size $7 \times 7$. This allows to see the context (and end) of the contact between two secondary structure elements (e.g. a contact between two $\beta$-strands). In other words, this choice of the window size and number of filters is motivated by the fact that information such as "$(i, j)$ is a contact between a $\beta$-strand at $i$ and a $\beta$-strand at $j$, the arrangement is antiparallel, the contact ends two residues after $i$ (and before $j$)" can be captured from a $7 \times 7$ window of the data, and well encoded in about 64 filters.

The third and final layer learns one filter (returning the predicted contact map) with the window size $9 \times 9$. Thus, the overall receptive window of the convolutional network is $15 \times 15$, which provides the required amount of context of the co-evolution data to predict the contacts. Particularly, the relative position (including the angle) between two contacting $\alpha$-helices can be well captured at this window size. At the same time, this deep architecture is different from having, say, a network with a single $15 \times 15$ convolutional layer because a non-deep network would require seeing many possible $15 \times 15$ configurations in a non-abstract manner, and would tend to generalize badly and overfit. In contrast, abstraction to higher-level features is provided by preceding layers in our architecture.

We used mean squared error loss, dropout 0.2 after input layer, 0.5 after each hidden layer, one pixel stride, no pooling. The network is trained in Lasagne (https://github.com/Lasagne) using the Adam algorithm [16] with learning rate 0.0001 for 100 epochs.

# 3 Results

To assess the performance of protein contact prediction methods, we have used the contact likeliness criterion for C$\beta$ distances (cf. Introduction), but the qualitative results are not dependent on the criterion chosen. We have evaluated predictions both in terms of Top 10 pairs that are predicted most likely to be in contact. It is estimated that in a protein one can observe $L$ to $3L$ contacts, where $L$ is the length of the amino acid chain. Thus we have also evaluated greater numbers of predicted contacts. We have assessed the predictions with respect to the sequence separation. It is widely accepted that it is more difficult to predict long-range contacts than the ones separated by few amino acids in the sequence space. At the same time, it is the long-range contacts that are most useful for restraining the protein structure prediction simulations [17]. Maintaining the order of amino acid types (feature vector construction method #1) yielded the best results in our case, which we focus on exclusively in the following.

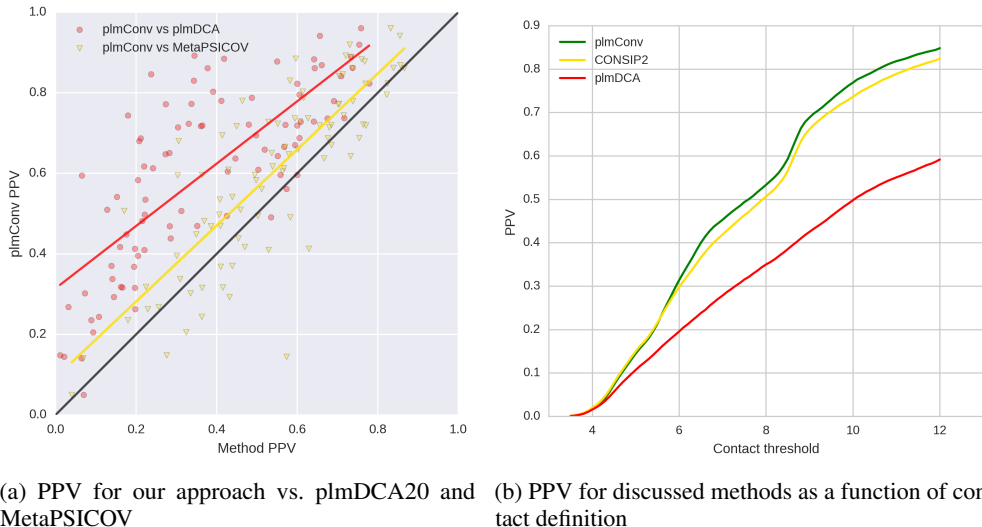

(a) PPV for our approach vs. plmDCA20 and MetaPSICOV

(b) PPV for discussed methods as a function of contact definition

Figure 2: Method performance. Panel (a): prediction accuracy of plmConv (Y-axis) vs plmDCA and MetaPSICOV (X-axis, in red and yellow, respectively); lines: least square fit, circles: individual comparisons. Panel (b): prediction accuracy, depending on contact definition. X-axis: C$\beta$ distance threshold for amino acid pair to be in contact.

**plmConv yields more accurate predictions than plmDCA**. We compared the predictive performance of the proposed plmConv method to plmDCA in terms of positive predictive value (PPV) at different prediction counts and different sequence separations (see Table 1 and Fig. 2a). Regardless of the chosen threshold, plmConv yields considerably higher accuracy. This effect is particularly important in context of long-range contacts, which tend to be underpredicted by plmDCA and related methods, but are readily recovered by plmConv. The notable improvement in predictive power is important, given that both plmDCA and plmConv use exactly the same data and same inference algorithm, but differ in the processing of the inferred co-evolution matrices. We posit that this may have longstanding implications for evolutionary coupling analysis, some of which we discuss below.

**plmConv is more accurate than MetaPSICOV, while remaining more flexible**. We compared our method to MetaPSICOV [18, 19], a method that performed best in the CASP11 experiment. We observed that plmConv results in overall higher prediction accuracy than MetaPSICOV (see Table 1 and Fig. 2a). This holds for all the criteria, except for the top-ranked short contacts. MetaPSICOV performs slightly better at the top-ranked short-range contacts, but they are easier to predict, and less useful for protein folding [17]. It is worth noting that MetaPSICOV achieves its high prediction accuracy by combining multiple sources of co-evolution data (including methods functionally identical to plmDCA) with predicted biophysical properties of a protein (e.g. secondary structure) and a feed-forward neural network. In plmConv we are able to achieve higher performance, by using (a) an arbitrary alignment and (b) a single co-evolution result, which potentially allows for tuning the hyperparameters of (a) and (b) to answer relevant biological questions.

| Separation | Method | Top 10 | $L/10$ | $L/5$ | $L/2$ | $L$ |
|---|---|---|---|---|---|---|
| | MetaPSICOV | 0.797 | 0.761 | 0.717 | 0.615 | 0.516 |
| All | plmDCA | 0.598 | 0.570 | 0.525 | 0.435 | 0.356 |
| | plmConv | **0.807** | **0.768** | **0.729** | **0.663** | **0.573** |
| | MetaPSICOV | **0.754** | **0.683** | **0.583** | 0.415 | 0.294 |
| Short | plmDCA | 0.497 | 0.415 | 0.318 | 0.229 | 0.178 |
| | plmConv | 0.724 | 0.654 | 0.581 | **0.438** | **0.320** |
| | MetaPSICOV | 0.710 | 0.645 | 0.559 | 0.419 | 0.302 |
| Medium | plmDCA | 0.506 | 0.438 | 0.355 | 0.253 | 0.180 |
| | plmConv | **0.744** | **0.673** | **0.583** | **0.428** | **0.304** |
| | MetaPSICOV | 0.594 | 0.562 | 0.522 | 0.436 | 0.339 |
| Long | plmDCA | 0.536 | 0.516 | 0.455 | 0.372 | 0.285 |
| | plmConv | **0.686** | **0.651** | **0.616** | **0.531** | **0.430** |

Table 1: Positive predictive value for all non-local (separation $6^+$ positions), short-range, mid-range and long-range ($6 - 11$, $12 - 23$ and $24^+$ positions) contacts. We demonstrate results for Top 10 contacts per protein, as well as customary thresholds of $L/10$, $L/5$, $L/2$ and $L$ contacts per protein, where $L$ is the length of the amino acid chain.

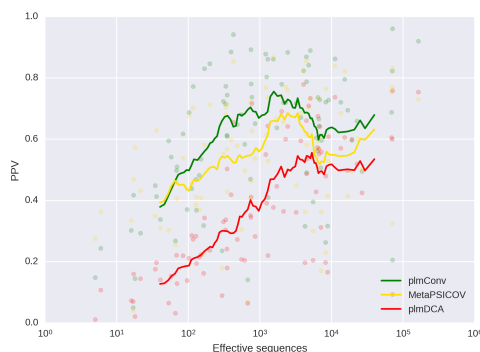

Figure 3: Positive predictive value for described methods at $L$ contacts considered as a function of the information content of the alignment. Scatter plot: observed raw values. Line plot: rolling average with window size 15.

**plmConv pushes the boundaries of inference with few sequences**. One of the major drawbacks of statistical inference for evolutionary analysis is its dependence on availability of high amounts of homologous sequences in multiple sequence alignments. Our method to a large extent alleviates this problem. As illustrated in Fig. 3, plmConv outperforms plmDCA accross all the range. MetaPSICOV appears to be slightly better at the low-count end of the spectrum, which we believe is due to the way MetaPSICOV augments the prediction process with additional data – a technique known to improve the prediction, that we have expressly *not* used in this work.

**plmConv predicts long-range contacts more accurately**. As mentioned above, it is the long-range contacts which are of most utility for protein structure prediction experiments. Table 1 demonstrates that plmConv is highly suitable for predicting long range contacts, yielding better performance across all the contact count thresholds.

**T0784: a success story**. One of the targets in CASP11 (target ID: T0784) was a DUF4425 family protein (BACOVA_05332) from *Bacteroides ovatus* (PDB ID: 4qey). The number of identifiable sequence homologs for this protein was relatively low, which resulted in uninterpretable contact map obtained by plmDCA. The same co-evolution statistics used as input to plmConv yielded a contact map which not only was devoid of the noise present in plmDCA's contact map, but also uncovered numerous long-range contacts that were not identifiable previously. The contact map produced by plmConv for this target is also of much higher utility than the one returned by MetaPSICOV. Note in Fig. 4c how MetaPSICOV prediction lacks nearly all the long-range contacts, which are present in the plmConv prediction.

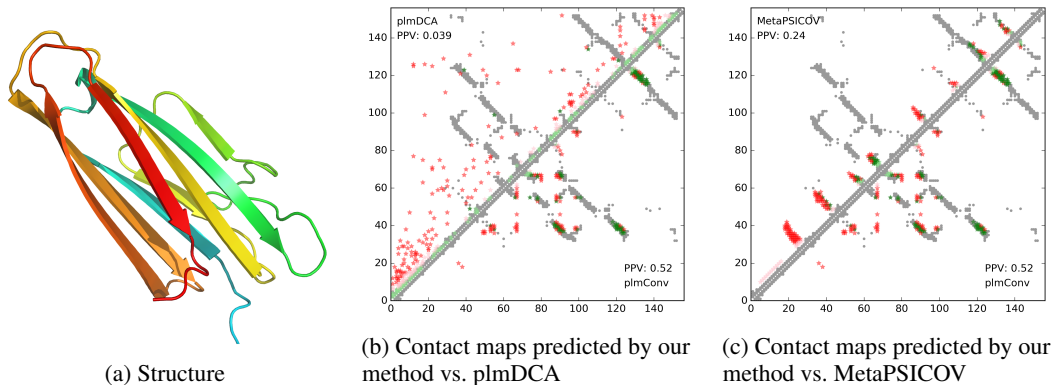

(a) Structure

(b) Contact maps predicted by our method vs. plmDCA

(c) Contact maps predicted by our method vs. MetaPSICOV

Figure 4: An example of one of CASP11 proteins (T0784), where plmConv is able to recover the contact map, which other methods cannot. True contacts (ground truth) marked in gray. Predictions of respective methods are marked in color, with true positives in green and false positives in red. Predictions along the diagonal with separation of 5 amino acids or less have not been considered in computing positive predictive value and have been marked in lighter colors in the plots.

## 4 Discussion and Conclusions

In this work we proposed an entirely new way to handle the outputs of the co-evolutionary analyses of multiple sequence alignments of homologous proteins. We demonstrated that this method is considerably superior to the current ways of handling the co-evolution data, able to extract more information from them, and consequently greatly aid protein contact prediction based on these data. Contact prediction with our method is more accurate and 2 to 3 times faster than with MetaPSICOV.

**Relevance to the field**. Until now, the utility of co-evolution-based contact prediction was limited because most of the proteins that had sufficiently high amount of sequence homologs had also their structures determined and available for comparative modelling. As plmConv is able to predict high-accuracy contact maps from as few as 100 sequences, it opens a whole new avenue of possibilities for the field. While there are only a few protein families that have thousands of known homologs but no known structure, there are hundreds which are potentially within the scope of this method. We postulate that this method should allow for computational elucidation of more structures, be it by means of pure computational simulation, or simulation guided by predicted contacts and sparse experimental restraints.

**plmConv allows for varying prediction parameters**. One of the strengths of the proposed method is that it is agnostic to the input data, in particular to the way input alignments are constructed and to the inference parameters (regularization strength). Therefore, one could envision using alignments of close homologs to elucidate the co-evolution of a variable region in the protein (e.g. variable regions of antibodies, extracellular loops of G protein–coupled receptors etc.), or distant homologs to yield structural insights into the overall fold of the protein. In the same way, one could vary the regularization strength of the inference, with stronger regularization allowing for more precise elucidation of the few couplings (and consequently contacts) that are most significant for protein stability or structure from the evolutionary point of view. Conversely, it is possible to relax the regularization strength and let the data speak for itself, which could potentially result in a better picture of the overall contact map and give a holistic insight into the evolutionary constraints on the structure of the protein in question.

The method we propose is directly applicable to a vast array of biological problems, being both accurate and flexible. It can use arbitrary input data and prediction parameters, which allows the end user to tailor it to answer pertinent biological questions. Most importantly, though, even if trained on the heavily constrained data set, it is able to produce results exceeding in predictive capabilities those of the state-of-the-art methods in protein contact prediction at a fraction of computational effort, making it perfectly suitable for large-scale analyses. We expect that the performance of the method will further improve when trained on a larger, more representative set of proteins.

**Acknowledgments**  Grant support: Deutsche Telekom Foundation, ERC Consolidator Grant "3DReloaded", ERC Starting Grant "VideoLearn".

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
