[Reviews · NeurIPS 2016]

Reviewer 1

Summary

The authors use convolutional neural network to predict protein contact maps from inferred evolutionary coupling between each pair of amino acids at any corresponding positions. The network, which architecture is, to some extent, biologically motivated, is trained using a limited set of proteins originating from CASP9 and CASP10 and tested on CASP11 (the 2010, 2012, and 2014 community-wide protein structure prediction challenges, respectively) proteins. It is shown to outperform both (a) plmDCA, which uses the same coupling matrices as input, but applies a more straightforward scoring function; and (b) CONSIP2, a best performer in the CASP11 experiment.

Qualitative Assessment

Overall, the paper is well written and the proposed method advances the state-of-the-art. However, the paper seems more suitable for publication in a computational biology venue than in NIPS as, from a machine learning perspective, it primarily uses a standard, off-the-shelf methodology. Specific comments 1. How are the bipartite and arbitrary graphs relevant to the proposed method? Seemingly, it does not utilize these concepts. 2. How many proteins are used for training, test? Are these all the proteins in the corresponding CASPs, subsets? 3. Fig 2a,b – perhaps highlight T078; is it a representative example? 4. It is claimed that “maintaining the order of amino acid types yielded the best results”, but is this actually demonstrated? 5. The authors conclude that plmConv “… is able to produce results exceeding in predictive capabilities those of the state-of-the-art methods … at a fraction of computational effort”; as computational effort is not explicitly discussed, it’s unclear to which methods this statement refers.

Confidence in this Review

2-Confident (read it all; understood it all reasonably well)


Reviewer 2

Summary

Predicting the contact map of a protein from the co-evolution of amino acid residues is an important problem in Bioinformatics (it has been found to improve 3-D structure prediction). This paper proposes a convolutional neural net to predict the contact map from the input of pairwise co-occurences of amino acid residues. The proposed method shows a substantial gain in accuracy over the state-of-the-art.

Qualitative Assessment

This paper is a nice application of convolutional neural nets to an important problem. The experimental results are convincing. 1. The authors propose 4 possibilities of constructing the feature vector but do not provide an evaluation of each. I assume they use option 1 in their current results ? 2. In table 1, what are the standard errors of the PPV? 3. Did the authors attempt to learn the hyperparameters e.g. field size, learning rate ? How did they arrive at their specific architecture ?

Confidence in this Review

3-Expert (read the paper in detail, know the area, quite certain of my opinion)


Reviewer 3

Summary

A novel application of convolutional neural networks to protein co-evolution matrixes is presented that performs significantly better than state of the art approaches for evolutionary conservation based contact prediction. This method has the potential to directly improve ab initio structure prediction, which is an important and unsolved problem in biophysics and machine learning.

Qualitative Assessment

Overall the paper presents a novel application of CNNs to co-evolution Protein Contact Prediction (PCP) with state-of-the-art results. The paper is a strong contribution to the field of coevolution PCP. The method has the potential to improve ab initio structure prediction accuracy, which could have wide ranging impacts on studying proteins without solved structural complexes. The paper itself contains flaws in the experimental setup, how the methods are described, and how the results are presented. Key issues: - Experimental setup: not enough unique training examples are used to compare to existing methods. More than 100K examples exist for this specific task, but only 307 are used. - A key related method (PconsC2) is missing from the experimental benchmarks. Method description: the model is not described in sufficient detail to be reproduced by a 3rd party with a reasonable level expertise. In particular, the loss function and model parameters are not sufficiently described. - Results presentation and interpretation: plots are hard to read because underlying data points are too lightly printed, too few examples are used for results comparisons, and several key claims are not sufficiently supported by data. Specific points - The number of examples used for training and testing is very small, especially when compared to numbers of examples used in related publications. For example, in the plmDCA paper, 10,000 examples are used, rather than the 307 in this paper. The authors claim they had trouble finding consistent homologue sets for all proteins, but there are exisiting pipelines for this task (e.g. the gremlin pipeline http://gremlin.bakerlab.org/). It would be especially interesting to see how the relative performance of the method changes when trained on a larger corpus of training examples, especially considering that PDB contains > 100K unique crystal structures that could each potentially be used as an example. - No justification is given for the binary contact/non-contact threshold of 8A (previous comparisons have used different cutoffs). It is likely that cutoffs at different resolutions will be useful for ab initio structure prediction. It would be interesting to see a comparison of pmlConv to plmDCA and CONSIP2 with different cutoff boundaries. It is potentially a concern that the cutoff was chosen because it maximizes the margin of plmConv over other methods, rather than because it corresponds to a physically important distance. - The model should also be benchmarked against PconsC2 (Skwark M., et al., “Improved Contact Predictions Using the Recognition of Protein Like Contact Patterns”, PLOS Computational Biology, 2014). This would be an especially informative comparison because PconsC2 uses an ensemble of linear classifiers for the exact same task (PCP from plmDCA matrixes). PconsC2 also includes learned spatial filters, which are missing from the current benchmarks. - The deep learning model is not adequately described to be reproducible. Especially important is that the paper does not mention what loss function is used for optimization (presumably binary cross entropy?) Also missing are descriptions of what convolutional strides and padding is used, what types of initialization are used (probably very important for learning on limited examples), whether batch norm is used, and whether dropout is used. - No interpretation is presented of what features the model learns. It would be interesting to apply a technique such as guided back propagation to input to provide an interpretation of which features are most important for classification of a given residue pair. - In figures 2 and 3, it is very hard to see the circles that represent individual data points. The circles should be presented with higher contrast so readers can better see the underlying data distribution, especially given that it is so sparse. - In figure 3, the presented data are very noisy. It would be worth reproducing this figure with more alignments across a larger set of proteins. - In figure 3, the x-axis is labeled “amount of sequences”, but the caption states the figure shows the “information content” of the alignment as the number of “non-redundant sequences”. It would be worthwhile to consider more informative metrics than the number of non-redundant sequences: an alignment in which each sequence has hamming distance exactly 1 from each other sequence is much less informative for co-evolution than an alignment with higher sequence divergence. A metric such as effective number of sequences (where sequences are weighted by normalized average hamming distance), or alignment entropy may be more informative summary statistics for this purpose. - The paper presents a semantic argument that larger layers could lead to overfitting, but could this be overcome through parameter regularization or dropout? This would allow convolutional filters to potentially detect longer range interactions. - The paper claims that plmConv yields higher accuracy than plmDCA for especially long range interactions. It would be interesting to plot PPV as a function of contact distance, since this figure would directly support this point (rather than the PPV table split by distance, which only partially supports this claim). - Section 2.4 proposes 4 schemes for featurizing the J matrix, but results are not reported for each. It would be informative to report how featurization scheme influences performance.

Confidence in this Review

3-Expert (read the paper in detail, know the area, quite certain of my opinion)


Reviewer 4

Summary

This paper proposes a new approach (plmConv) to perform contact prediction which they show to be superior compared to state of the art methods in extracting information from co-evolution data of amino acids. Approach: They represent the Co-evolution statistics as a L x L x 21 x 21 matrix. In order to predict the contact map C i,j between the 2 residues i and j, from the co-evolution statistics J i,j,k,l they model J as a 3-D image by concatenating the 21 x 21 dimension into a 441 vector, and apply the required padding to the location dimensions. They model a small CNN with 3 convolutional layers. The first layer has 128 filters of size 1 x 1, and the second has 64 filters of size 7x7 and the 3rd has 1 filter of size 9x9. Strengths: 1. The paper overall is well written- - They introduce the problem and clearly illustrate why that is an important problem in their field, making it an interesting read for people outside their research area. Although I feel that section 2 can be organized in a better way. 2. I like the way they modeled the problem using Convolutional Neural Networks, giving clear justification for why a CNN would do a good job. 3. Careful network design - The number of filters and filter sizes are carefully chosen for each layer and it makes sense for their problem. 4. Analysis of what each layer learned to capture. - For example they mention how Cysteine has strong coupling with Histidine and this is reflected in the values learned by the network. 5. The proposed approach more accurately predicts contacts when compared to plmDCA (a inverse Potts model), and CONSIP2 6. Their method plmConv can predict accurate contact maps from as few as 100 sequences. 7. Ability to predict long-range contacts significantly accurately compared to other methods. 8. Their approach was able to obtain a high quality contact map of Target ID T 0784, with contacts which was previously not identifiable. 9. Generic method; It can be applied to a wide variety of problems which have some kind of spatial structure 10. Computationally effective; plmConv is extremely fast compared to previous state of the art methods and has the capacity to further improve when trained on larger, more representative set of proteins.

Qualitative Assessment

I feel that this paper is worth being accepted as an oral talk. My decision is based on the fact that they have modelled the problem well, and designed the filters appropriately. Also their novel approach beats the state of the art models by a huge margin when just trained with 100 samples, and it is pretty convincing to me that their model has the capacity to learn more complicated information, when provided with more data. Also their approach seems directly applicable to several problems in the field. Although, I feel that the authors should address the following points: 1. The argument they make in L201 - L206 about designing filter size for predicting the contact between beta strands can be made stronger if it can be shown that the network also learned to predict the same. Same applies to the next paragraph on the filters designed as to predict the relative position and angle between 2 alpha helices. 2. Any References pointing to the fact that long-range contacts are more useful for restraining the protein structure prediction, would be good to site. Author response: In the rebuttal, the authors mentioned that they will be citing this paper doi:10.1016/S0301-4622(99)00010-1 stating the importance of long range contacts in the final version. Clarity: 3. Section 2 can be written more clearly. It is a mixture of models, datasets and approaches. It is not clear to me which ones are theirs and which ones are being compared to. Author response: In the Rebuttal, the authors mentioned that they will improve the structure and formulations of Section 2 4. Section 2.4 L139 - L142 It is not clear which of the 4 settings led to the best results. It would be great if the authors can address the above weakness (mainly 1,3, and 4) in the rebuttal.

Confidence in this Review

2-Confident (read it all; understood it all reasonably well)


Reviewer 5

Summary

In this paper, a novel method for protein contact prediction is proposed. The method is based on a version of convolutional neural networks capable to process input grids where each element is a bipartite weighted graph. The weights are given by the co-evolution statistics of the amino acids (considering relative positions in the input sequences). The approach is very interesting and also promising in terms of experimental results. I suggest to accept this paper as it is.

Qualitative Assessment

Very good in all respects

Confidence in this Review

2-Confident (read it all; understood it all reasonably well)


Reviewer 6

Summary

The authors propose a method based on pseudolikelihood maximization and convolution networks for protein contact map prediction. The proposed method generates multiple sequence alignment results using jackhmmer and uses plmDCA to obtain inferred evolutionary couplings in an L x L x 21 x 21 array. Then, the array is used as input to the convolutional neural network to predict L x L contact map. This method achieves better PPV than the current state-of-the-art alternatives (plmDCA and CONSIP2).

Qualitative Assessment

While the paper has its own merits, unfortunately it has several issues that need to be addressed. A. The authors stressed on several occasions that the proposed method allows great flexibility with regard to the input data (L13, L145-155, L296-032). However, I am not entirely sure that the manuscript provides enough rationale for the argument. Since the proposed method is based on pseudolikelihood maximization of co-evolution statistics, I think the input data for the proposed method must have similar input data type, which limits its flexibility. On the other hand, while CNN provides some flexibility to the proposed method, it’s difficult to fully consider it as the authors’ contribution. Although the input data have some different characteristics (L153-155), still CNN has already been widely used on Graph-Structured data as mentioned in the manuscript. B. Are there specific reasons or advantages for considering co-evolution statistics input as graph-valued images? Does CNN exploit any characteristics of graph-valued images? If not, why the authors consider the input data as graph-valued images rather than just thinking the method as applying CNN to matrix type input data. C. In L136-144, authors stated that four possibilities to construct feature vector would be evaluated, and in L274, the authors stated, “maintaining the order of amino acid types yielded the best results, as shown above.” However, I couldn’t find the corresponding contents and results comparison in the results section. D. The neural network architecture description in section2.7 is somewhat confusing and needs more details. In L190-191, the authors stated “The first layer learns 128 filters of size 1x1. Thus 441 input features are compressed to 128 learned features.” Are these correctly stated as the authors’ intention? According to the manuscript, I think the input feature is 21x21 matrix and 128 filters of size 1x1 generate 128 learned features of size 21x21 (assuming 1 stride and no pooling). Then the features are not compressed, and rather decompressed. In addition, the authors’ did not state enough details for reproducibility (i.e., pooling layer, stride size, used library for the implementation). E. Did the authors perform experiment of compared tools (plmDCA, CONSIP2) as well? Or were the results excerpted from the existing research? Either way it would be better to clarify the details in the experiment. If it’s the former, I wonder if CONSIP2 was trained with CASP9 and CASP10, and tested with CASP11 as well. Additionally the results of CONSIP2 show some differences from the existing research (Monastyrskyy, Bohdan, et al. "New encouraging developments in contact prediction: Assessment of the CASP11 results." Proteins: Structure, Function, and Bioinformatics (2015).). Are there any differences in the experiment setting? F. The authors only used PPV as evaluation metric. While the proposed method clearly has its merits in terms of PPV, Figure5(C) suggests that it may still have low sensitivity. To prevent over optimistic view of the proposed method, I think the authors should also evaluate the proposed metric with other metrics (i.e., sensitivity, MCC, g-mean, F-measure)

Confidence in this Review

2-Confident (read it all; understood it all reasonably well)